# Quantum Pumping with Adiabatically Modulated Barriers in Three-Band Pseudospin-1 Dirac–Weyl Systems

**DOI:** 10.3390/e21020209

**Published:** 2019-02-22

**Authors:** Xiaomei Chen, Rui Zhu

**Affiliations:** Department of Physics, South China University of Technology, Guangzhou 510641, China

**Keywords:** quantum transport

## Abstract

In this work, pumped currents of the adiabatically-driven double-barrier structure based on the pseudospin-1 Dirac–Weyl fermions are studied. As a result of the three-band dispersion and hence the unique properties of pseudospin-1 Dirac–Weyl quasiparticles, sharp current-direction reversal is found at certain parameter settings especially at the Dirac point of the band structure, where apexes of the two cones touch at the flat band. Such a behavior can be interpreted consistently by the Berry phase of the scattering matrix and the classical turnstile mechanism.

## 1. Introduction

After quantized particle transport driven by adiabatic cyclic potential variation was proposed by D. J. Thouless in 1983 [1], such a concept has attracted unceasing interest among researchers concerning its theoretical meaning and potential applications in various fields such as a precision current standard and neural networks [2,3,4]. Mechanism of the quantum pump can be interpreted consistently by the Berry phase of the scattering matrix in the parameter space within the modulation cycle [2] and the classic turnstile picture [5,6]. Usually, the pumped current is unidirectional when the phase difference between the two driving parameters is fixed. In the turnstile picture, the opening order of the two gates is defined by the driving phase. The first-opened gate let in the particle and the second-opened gate let it out forming a direct current (DC) current after a cycle is completed. However, reversed DC current direction has been discovered in various systems even when the driving phase is fixed such as in monolayer graphene [6] and carbon nanotube-superconductor hybrid systems [7]. This is because that conventionally a “gate” is defined by a potential barrier and higher barriers allow smaller transmission probabilities. However, as a result of the Klein tunneling effect, the potential barrier becomes transparent regardless of its height at certain parameter settings. When higher barrier allows even stronger transmission, the opening and closing of a “gate” in the quantum pump is reversed and so the driven current is reversed with the driving phase difference unchanged. The same phenomenon is also discovered in the superconductive carbon nanotube when Andreev reflection again violates the higher-barrier-lower-transmission convention and reversed the pumped current under the same driving forces. This turnstile interpretation of the reversed pumped current coincides with the Berry phase of the scattering matrix in the parameter space within the modulation cycle. However, a clear comparison between the two mechanisms is lacking, which is one of the motivations of this work.

About the significance of the comparison between the Berry phase picture and the classic turnstile mechanism of the adiabatic quantum pumping, we would like to make some further background remarks.

The classic turnstile mechanism and the Berry-phase-of-scattering-matrix picture of adiabatic quantum pumping are proposed based on different physical origin. The former is from classic mechanics and the latter is from quantum mechanics. General agreement between them is certainly a surprising result because they have at least the following differences on the conceptual level.

(1) In quantum mechanics, the leftward and rightward transmission probabilities of both the symmetric and asymmetric double-barrier structure (The former means the height and width of the two barriers are exactly the same. The latter means the height and width of the two barriers are different.) are exactly the same if a typical two-lead device is considered. The difference between the leftward and rightward transmission is in the phase factor of the transmission amplitudes. Such a phase difference gives rise to a nontrivial Berry phase formed by cyclic modulation of the two barriers with $V_{1}=V_{1\omega }\cos\left(\omega t+\varphi \right)$ and $V_{2}=V_{2\omega }\cos\left(\omega t\right)$ when time-reversal symmetry is not conserved such as excluding φ=0 or π. In the turnstile picture, the two barriers are treated separately like two gates. The opening and closing of the two gates is determined by the transmission probability of the corresponding barrier potential. When higher barrier generates smaller transmission probability, the opening and closing of the gate is defined conventionally: lifting the barrier means closing the gate and lowering the barrier means opening the gate. Because the charge carrier density in a typical semiconductor can be up to $10^{10}∼10^{15}/{\mathrm{cm}}^{3}$ at room temperature, only a small change in the barrier height and hence in the transmission probability can justify the definition of the “opening” and “closing” of the gate for charge carriers. While the same gate-modulation is applied, charge carriers are driven unidirectionally to one of the reservoirs like the turnstile in daily life. No phase factor is involved in the picture at all.

(2) In the quantum interpretation of parametric pump, time-reversal symmetry is a vital factor. The most prominent case is when the driving phase difference φ=π. In this case, time-reversal symmetry is conserved as the two parameters vary periodically. Because of this, a DC current is forbidden even when the classic turnstile gives rise to the largest mass flow when the phase lag is the largest. At this point, the classic and quantum models become incomparable, which is out of our present discussion.

Therefore, we feel a confirmation of the agreement between the two mechanisms is a significant step forward to understand the underlying physics of adiabatic quantum pumping. In preparation of this work, we have proved it in various parameter settings in different systems such as two-dimensional electron gas and graphene besides the present pseudospin-1 Dirac–Weyl system by calculating term by term Equation (Equation 10). Although we could not provide a general proof, up to now, no numerical evaluation violates such a conclusion.

After the idea of the adiabatic quantum pump (also called Thouless pump and parametric pump) is proposed, such a mechanism has been investigated in various transport devices such as a single spin in diamond [8], quantum-dot structures [9], Rashba nanowires [10], Mach–Zehnder interferometers [11], the magnetic nanowire with double domain walls [12], magnetic-barrier-modulated two dimensional electron gas [13], mesoscopic rings with Aharonov–Casher and Aharonov–Bohm effect [14], magnetic tunnel junctions [15], and monolayer graphene [6,16,17]. Correspondingly, theoretical techniques have been put forward for the treatment of the quantum pumps such as the scattering matrix formalism [18], non-equilibrium Green’s function [19,20,21,22], and the quantum master equation approach [9]. In this work, we use the scattering matrix approach for alternating current (AC) transport, which defines the Berry phase formed within the looped trajectory of the two varying parameters [2,18,23].

Recently, after realization of the monolayer graphene, which is characterized as a pseudospin-1/2 Dirac–Weyl fermionic material, a family of general pseudospin-*s* (s=1/2,1,3/2,⋯) Dirac–Weyl fermionic materials has been proposed by sharing similar band structure with one or several pairs of Dirac cones. Pseudospin-1 materials with a band structure of two Dirac cones and a flat band through where the cones intersect have attracted intense interest in the physical society currently. Numerical or experimental studies have proposed various host materials of such band structure such as conventional crystal with special space group symmetries [24,25], in the electronic, photonic, and phononic Lieb lattice [26,27,28,29,30,31,32,33], kagome lattice [31,34], dice or $T_{3}$ lattice [31,35,36,37,38,39,40,41,42,43,44,45], and $K_{4}$ crystal [46]. Along with these progress in material building, various transport properties of the pseudospin-1 Dirac–Weyl fermions have been investigated such as super Klein tunneling effect [47,48,49], magneto-optics [50], Hall quantization [51], and Hofstadter butterfly [52] in a magnetic field. While the adiabatic quantum pumping process serves as an important platform to detect various properties of novel quantum states, it is worthwhile to apply the idea on newly-emerged pseudospin-1 Dirac–Weyl materials. To understand how their particular transport properties modify the adiabatically-driven pumped current is the other motivation of this work.

The plan of the present work is as follows. In Section 2, the model is introduced and the key formulas for the scattering matrix, Berry phase, and pumped current are given. In Section 3, we present numerical results of the pumped current and discussions of the underlying mechanisms. In Section 4, a rigorous proof of the consistency between the quantum Berry phase picture and the classic turnstile mechanism for adiabatic quantum pumping is provided. A brief summary is given in Section 5. Detailed derivation of the boundary condition and the scattering matrix are provided in Appendix A and Appendix B, respectively.

## 2. Model and Formalism

We consider a two-dimensional (2D) non-interacting pseudospin-1 Dirac–Weyl system modulated by two time-dependent electric potential barriers illustrated in Figure 1. The pseudospin-1 Dirac–Weyl fermions are charged quasiparticles originating from free electrons moving in the three-band structure consisting of gapless tip-to-tip two cones intersected by a flat band, which is shown in Figure 1c. Their dynamics are governed by the dot product of the spin-1 operator and the momentum. Matrices of the spin-1 operator $\hat{S}=\left({\hat{S}}_{x},{\hat{S}}_{y},{\hat{S}}_{z}\right)$ in the ${\hat{S}}_{z}$-representation (the representation that ${\hat{S}}_{z}$ is diagonalized) can be deduced from spin-lifting/lowering operators ${\hat{S}}_{\pm }={\hat{S}}_{x}\pm {\hat{S}}_{y}$ by ${\hat{S}}_{\pm }|S,S_{z}\rangle =\sqrt{\left(S\mp S_{z}\right)\left(S\pm S_{z}+1\right)}|S,S_{z}\pm 1\rangle$ [53]. Simple algebra leads to the results that(1)$$\begin{array}{ccc} {\hat{S}}_{x}=\frac{1}{\sqrt{2}}\left(\begin{array}{ccc} 0 & 1 & 0\\ 1 & 0 & 1\\ 0 & 1 & 0 \end{array}\right), & {\hat{S}}_{y}=\frac{1}{\sqrt{2}}\left(\begin{array}{ccc} 0 & -i & 0\\ i & 0 & -i\\ 0 & i & 0 \end{array}\right), & {\hat{S}}_{z}=\left(\begin{array}{ccc} 1 & 0 & 0\\ 0 & 0 & 0\\ 0 & 0 & -1 \end{array}\right). \end{array}$$

By applying AC gate voltages, Hamiltonian of the pseudospin-1 Dirac–Weyl fermions has the form(2)$$\hat{H}=-i\hbar v_{g}\hat{S}\cdot \nabla +V\left(x,t\right),$$
where $\hat{S}$ is the spin-1 operator defined in Equation (Equation 1), $v_{g}\approx 10^{6}$ m/s is the group velocity associated with the slope of the Dirac cone. As shown in Figure 1a, the potential function has the form(3)$$V\left(x,t\right)=\left\{\begin{array}{cc} V_{0}+V_{1}\left(t\right), & 0<x<L_{1},\\ V_{0}+V_{2}\left(t\right), & L_{2}<x<L_{3},\\ 0, & \mathrm{others}, \end{array}\right.$$
with $V_{1}\left(t\right)=V_{1\omega }\cos\left(\omega t+\varphi \right)$ and $V_{2}\left(t\right)=V_{2\omega }\cos\left(\omega t\right)$. The Fermi energy of the two reservoirs to the two sides of the double-barrier structure are equalized to eliminate the external bias and secure energy-conserved tunneling. While the frequency of the potential modulation ω is small compared to the carrier interaction time (Wigner delay time) with the conductor, the quantum pump can be considered “adiabatic” [1,18,23]. In this case, one can employ an instant scattering matrix approach, which depends only parametrically on the time *t*. The Wigner–Smith delay time can be evaluated by $\tau =\mathrm{Tr}(-i\hbar s^{†}\frac{\partial s}{\partial E_{F}})$, with s the scattering matrix defined in Equation (Equation 5). Calculations below show $\tau \approx 10^{-14}$ s for all the parameter values. Thus, the adiabatic condition can be well justified when ω is in the order of MHz [3].

For studying the transport properties, the flux normalized scattering modes in different regions can be expressed in terms of the eigenspinors as(4)$$\Psi =\left(\begin{array}{c} \psi_{1}\\ \psi_{2}\\ \psi_{3} \end{array}\right)=\left\{\begin{array}{cc} a_{l}\Psi_{\to }+b_{l}\Psi_{\leftarrow }, & x<0,\\ a_{1}\Psi_{1\to }+b_{1}\Psi_{1\leftarrow }, & 0<x<L_{1},\\ a_{2}\Psi_{\to }+b_{2}\Psi_{\leftarrow }, & L_{1}<x<L_{2},\\ a_{3}\Psi_{2\to }+b_{3}\Psi_{2\leftarrow }, & L_{2}<x<L_{3},\\ a_{r}\Psi_{\leftarrow }+b_{r}\Psi_{\to }, & x>L_{3}, \end{array}\right.$$
where $k_{x}=\sqrt{E_{F}^{2}/{\left(\hbar v_{g}\right)}^{2}-k_{y}^{2}}$ with $E_{F}$ the quasiparticle energy at the Fermi level of the reservoirs. $\Psi_{\to }=\frac{1}{2\sqrt{\cos\theta }}{\left(e^{-i\theta },\sqrt{2}s,e^{i\theta }\right)}^{T}e^{ik_{x}x}$ for $E_{F}\neq 0$ (quasiparticles on the two cone bands) and $\frac{1}{\sqrt{2}}{\left(-e^{-i\theta },0,e^{i\theta }\right)}^{T}$ (we also identify it as $\Psi_{0}$ for discussions in the next section) for $E_{F}=0$ (quasiparticles on the flat band). $\theta =\arctan(k_{y}/k_{x})$, and $s=\mathrm{sgn}(E_{F})$. $\Psi_{\leftarrow }$ can be obtained by replacing $k_{x}$ with $-k_{x}$ in $\Psi_{\to }$; $\Psi_{i\to }$/$\Psi_{i\leftarrow }$ (i=1,2) can be obtained by replacing $k_{x}$ with $q_{xi}=\sqrt{{\left(E_{F}-V_{0}-V_{i}\right)}^{2}/{\left(\hbar v_{g}\right)}^{2}-k_{y}^{2}}$ and *s* with $s_{i}^{\prime }=\mathrm{sgn}\left(E_{F}-V_{0}-V_{i}\right)$ in $\Psi_{\to }$/$\Psi_{\leftarrow }$. The flux normalization factor $2\sqrt{\cos\theta }$ is obtained [55] by letting $\Psi^{†}\left(\partial \hat{H}/\partial k_{x}\right)\Psi =1$. $\psi_{i}$ (i=1,2,3) picks up the *i*-th row of the spinor wave function in all the five regions. Note that quasiparticles on the flat band contribute no flux in the *x*-direction. However, it must be taken into account in the pumping mechanisms while the Fermi energy lies close to the Dirac point. We will go to this point again in the next section.

The boundary conditions are that $\psi_{1}+\psi_{3}$ and $\psi_{2}$ are continuous at the interfaces, respectively [48]. The derivation of the boundary condition is provided in Appendix A. After some algebra, the instant scattering matrix connecting the incident and outgoing modes can be expressed as(5)$$\left(\begin{array}{c} b_{l}\\ b_{r} \end{array}\right)=\left(\begin{array}{cc} r & t^{\prime }\\ t & r^{\prime } \end{array}\right)\left(\begin{array}{c} a_{l}\\ a_{r} \end{array}\right)=s\left(V_{1},V_{2}\right)\left(\begin{array}{c} a_{l}\\ a_{r} \end{array}\right),$$
where s is parameter-dependent. Detailed derivation of the scattering matrix is provided in Appendix B.

The DC pumped current flowing from the α reservoir at zero temperature could be expressed in terms of the Berry phase of the scattering matrix formed within the looped trajectory of the two varying parameters as [2,18,23](6)$$I_{p\alpha }=\frac{\omega e}{2\pi }\int_{A}\Omega \left(\alpha \right)dV_{1}dV_{2},$$
where(7)Ωα=∑βIm∂sαβ*∂V1∂sαβ∂V2.
*A* is the enclosed area in the $V_{1}$-$V_{2}$ parameter space. While the driving amplitude is small ($V_{i\omega }\ll V_{0}$), the Berry curvature can be considered uniform within *A* and we have(8)$$I_{p\alpha }=\frac{\omega e\sin\varphi V_{1\omega }V_{2\omega }}{2\pi }\Omega \left(\alpha \right).$$
Conservation of current flux secures that the pumped currents flowing from the left and right reservoirs are equal: $I_{pl}=I_{pr}$. The angle-averaged pumped current can be obtained as(9)$$I_{p\alpha T}=\int_{-\pi /2}^{\pi /2}I_{p\alpha }\cos\theta d\theta .$$

## 3. Results and Discussion

Previously, we know that transport properties of the pseudospin-1 Dirac–Weyl fermions differs from free electrons in two ways. One is super Klein tunneling, which gives perfect transmission through a potential barrier for all incident angles while the quasiparticle energy equals one half the barrier height [48]. The other is particle-hole symmetry above and below the Dirac point of a potential barrier, which is a shared property with pseudospin-1/2 Dirac–Weyl fermions on monolayer graphene [56]. It gives that the transmission probability closely above and below the Dirac point is mirror symmetric because hole states with identical dispersion to electrons exist within the potential barrier unlike the potential barrier formed by the energy gap in semiconductor heterostructures. These two properties are demonstrated in the conductivity through a single potential barrier shown in Figure 1d. As a result of super Klein tunneling and because the conductivity also depends on the velocity or Fermi wavevector of the charge carriers, the maximum is parabolically-shaped under the present parameter settings and occurs at the Fermi energy larger than half the barrier height. For higher potential barriers, the maximum can be a sharp Λ-shaped peak appearing at the Fermi energy equal to half the barrier height [57]. Because of the existence of the local maximum peak and the V-shape local minimum in the single-barrier transmission probability and hence in the conductivity, it occurs that under certain conditions higher barrier allows larger quasiparticle transmission. The mechanisms of an adiabatic quantum pump in a mesoscopic system can be illustrated consistently by a classic turnstile picture and by the the Berry phase of the scattering matrix in the parameter space [2,5,6]. The turnstile picture can be illustrated within the framework of the single electron approximation and coherent tunneling constrained by the Pauli principle. The two oscillating potential barriers work like two “gates" in a real turnstile. Usually, lower potential allows larger transmissivity and thus defines the opening of one gate. When the two potentials oscillate with a phase difference, the two gates open one by one. Constrained by the Pauli principle, only one electron can occupy the inner single-particle state confined in the quantum well formed by the two potential barriers at one time, electrons flow in a direction determined by the driving phase difference. However, in monolayer graphene and in the pseudospin-1 Dirac–Weyl system, Klein tunneling, super Klein tunneling, and particle-hole symmetry at the Dirac point give rise to a reversal of the transmissivity-barrier height relation. As a result, the direction of the DC pumped current is reversed.

Numerical results of the pumped current are shown in Figure 2. It can be seen from Figure 1d that when the value of $E_{F}$ is between 70 meV and 100 meV, conductivity through higher potential barriers is larger than that through lower barriers. Angular dependence of the pumped current at Fermi energies selected within this range is shown in Figure 2b. With φ fixed at π/2, potential barrier $V_{1}$ starts lowering first and then it rises and $V_{2}$ starts lowering. Usually (like in a semiconductor heterostructure), higher potential barriers give rise to smaller transmission probability. The process can be interpreted as “gate” $V_{1}$ “opens” first allowing one particle to enter the middle single-particle state from the left reservoir and then it “closes” and “gate” $V_{2}$ “opens” allowing the particle to leave the device and enter the right reservoir. This completes a pump cycle and a DC current is generated. Such is the classical turnstile picture of the pumping mechanism. However, for pseudospin-1 Dirac–Weyl fermions, higher potential barriers give rise to larger transmission probability under certain parameter settings as demonstrated in Figure 1d. In the classical turnstile picture, this means that the definition of “opening” and “closing” of the "gate" is reversed. This is the reason for the negative (direction-reversed) pumped current shown in Figure 2b.

It can also be seen in Figure 2 that this turnstile picture of quantum pumping works for all parameter settings by comparing with Figure 1d. As a result of particle-hole symmetry above and below the Dirac point of a potential barrier, transmission probability of the pseudospin-1 Dirac–Weyl fermions demonstrate a sharp *V*-shape local minimum at the Dirac point. It should be noted that, at the Dirac point, eigenspinor wavefunction of the Hamiltonian is $\Psi_{0}$ and the transmission probability is exactly zero. We singled out this point in all of our calculations. Below the Dirac point, higher potential barriers allow larger transmission probability. Above the Dirac point, higher potential barriers allow smaller transmission probability. In addition, the difference is very sharp giving rise to a sharp negative pumped current below the Dirac point and a sharp positive pumped current above the Dirac point as shown in Figure 2d. In vast Fermi energy regime, the pumped DC current flows in the same direction for all incident angles as shown in panels (a), (b), and (c) of Figure 2, giving rise to smooth angle-averaged pumped current shown in Figure 2d. It should also be noted that the sharp current peak close to the Dirac point does not diverge and the current has an exact zero value at the Dirac point by taking into account quasiparticles on the flat band, which is a stationary state while the wavevector in the pump–current direction (*x*-direction in Figure 1a) is imaginary. The finite value of the pump–current peak is shown in the zoom-in inset of Figure 2d.

The previous discussion is based on the classical turnstile mechanism, while the pumped current is evaluated by the Berry phase of the scattering matrix formed from the parameter variation with a looped trajectory (Equation (Equation 8)). Such a consistency needs further looking into, which is elucidated in the next section.

## 4. Consistency between the Turnstile Model and the Berry Phase Treatment

In previous literature, consistency between the turnstile model and the Berry phase treatment is discovered while a clear interpretation is lacking.

Berry curvature Ω(α) of the scattering matrix s is defined by Equation (Equation 7) with s defined in Equation (Equation 5). *t*/$t^{\prime }$ and *r*/$r^{\prime }$ are the transmission and reflection amplitudes generated by incidence from the left/right reservoir with $t^{\prime }=t$ and $r^{\prime }=-r^{*}t/t^{*}$.

Without losing generosity, we consider a conductor modulated by two oscillating potential barriers $X_{1}=V_{1}$ and $X_{2}=V_{2}$ with the same width and equilibrium height. By defining the modulus and argument of *t* and *r* as $t=\rho_{t}e^{i\phi_{t}}$ and $r=\rho_{r}e^{i\phi_{r}}$, we have(10)$$\Omega \left(l\right)=\sum_{i=t,r}\rho_{i}\frac{d\rho_{i}}{dV_{1}}\frac{d\phi_{i}}{dV_{2}}-\rho_{i}\frac{d\phi_{i}}{dV_{1}}\frac{d\rho_{i}}{dV_{2}}.$$
Analytic dependence of $\rho_{i}$ and $\phi_{i}$ on the parameters $V_{1}$ and $V_{2}$ cannot be explicitly expressed. We show numerical results of the Berry curvature $\Omega \left(l\right)$ and the eight partial derivatives on the right-hand side of Equation (Equation 10) in Figure 3. For convenience of discussion, the parameter space in Figure 3a to (i) is divided into four blocks. It can be seen from Figure 3a that $\Omega \left(l\right)$ is negative in block II, positive in block III, and nearly zero in blocks I and IV. For the term $\rho_{t}\frac{d\rho_{t}}{dV_{1}}\frac{d\phi_{t}}{dV_{2}}-\rho_{t}\frac{d\phi_{t}}{dV_{1}}\frac{d\rho_{t}}{dV_{2}}$ in Equation (Equation 10), $\rho_{t}>0$, $\frac{d\phi_{t}}{dV_{2}}\approx \frac{d\phi_{t}}{dV_{1}}$ is negative throughout the four blocks and $\frac{d\rho_{t}}{dV_{1}}\approx \frac{d\rho_{t}}{dV_{2}}$ is positive in block II and negative in block III (see Figure 3b,g). As a result, this term approximates zero in blocks II and III. For the term $\rho_{r}\frac{d\rho_{r}}{dV_{1}}\frac{d\phi_{r}}{dV_{2}}-\rho_{r}\frac{d\phi_{r}}{dV_{1}}\frac{d\rho_{r}}{dV_{2}}$ in Equation (Equation 10), $\rho_{r}>0$, $\frac{d\phi_{r}}{dV_{2}}-\frac{d\phi_{r}}{dV_{1}}$ is positive throughout the four blocks (see Figure 3h,i), $\frac{d\rho_{r}}{dV_{1}}\approx \frac{d\rho_{r}}{dV_{2}}$ is positive in block III and negative in block II. It can also be seen from Figure 3 that in blocks I and IV the values of the two terms cancel out each other giving rise to nearly zero $\Omega \left(l\right)$. Therefore, the combined result of the two terms is that $\Omega \left(l\right)>0$ when $\frac{d\rho_{r}}{dV_{1}}>0$ and $\Omega \left(l\right)<0$ when $\frac{d\rho_{r}}{dV_{1}}<0$. This means that the Berry phase is positive and hence the pumped current is positive when higher potential barrier allows larger reflection probability in block III and that the Berry phase is negative and hence the pump–current direction is reversed when higher potential barrier allows smaller reflection probability in block II. Because $\rho_{r}^{2}+\rho_{t}^{2}=1$, larger reflection probability means smaller transmission probability, and consistency between the Berry phase picture and the classical turnstile model is numerically proved in the pseudospin-1 Dirac–Weyl system.

If we consider normal incidence, consistency between the Berry phase picture and the classic turnstile model becomes straightforward. For normal incidence, derivative of *t*/*r* with respect to $V_{2}$ is equal to derivative of $t^{\prime }$/$r^{\prime }$ with respect to $V_{1}$. Hence, we have(11)$$\Omega \left(l\right)=2\rho_{r}\frac{d\rho_{r}}{dV_{1}}\left(\frac{d\phi_{t}}{dV_{1}}-\frac{d\phi_{r}}{dV_{1}}\right).$$

From Figure 3, we can see that $\frac{d\phi_{t}}{dV_{1}}-\frac{d\phi_{r}}{dV_{1}}$ is positive throughout the parameter space. Therefore, the Berry phase has the same sign with $\frac{d\rho_{r}}{dV_{1}}$, which demonstrates consistency between the Berry phase picture and the classic turnstile mechanism of the adiabatic quantum pumping.

Besides data shown in Figure 3, which are contours of the Berry curvature $\Omega \left(l\right)$ and the eight derivatives on the right-hand side of Equation (Equation 10) in the parameter window of ±1 meV for both $V_{1}$ and $V_{2}$ with $V_{0}=100$ meV, $E_{F}=100$ meV, and θ=0.5 in radians, we have numerically targeted dozens more parameter windows of ±1 meV for $V_{1}$ and $V_{2}$ at other values of $V_{0}$, $E_{F}$, and θ and no obtained results violate consistency of the two mechanisms. Although the main focus of the present manuscript is the pseudospin-1 Dirac–Weyl fermions, we numerically confirmed consistency between the two mechanisms in various parameter settings in different systems such as two-dimensional electron gas, graphene, and the pseudospin-1 Dirac–Weyl system by calculating term by term Equation (Equation 10). Although we could not provide a general proof, up to now, no numerical evaluation violates such a conclusion.

We observe in this work and previously that the pump–current direction can be reversed in systems with linear bands such as graphene and pseudospin-1 Dirac–Weyl system. Up to now, similar behavior has not been observed in systems with parabolic band dispersion such as in the semiconductor two-dimensional electron gas. We remark that a quantitative argument of the underlying reason for the dependence of the adiabatic quantum pumping behavior on the band structure as observed numerically is lacking, due to the topological difference of the band structure.

## 5. Conclusions

In summary, adiabatic quantum pumping in a periodically modulated pseudospin-1 Dirac–Weyl system is studied. By using two AC electric gate-potentials as the driving parameters, a direction-reversed pumped current is found by the Berry phase of the scattering matrix at certain parameter regimes as a result of super Klein tunneling and particle-hole symmetry close to the Dirac point of the band structure. Such a phenomenon originates from the abnormal transmission behavior of the Dirac–Weyl quasiparticles that sometimes they transmit more through a higher electric potential barrier. As a result, definition of the “opening" and “closing" of a gate is reversed in the classic turnstile picture and hence direction of the pumped DC current is reversed. We also provide rigorous proof of the consistency between the quantum Berry phase picture and the classic turnstile mechanism.

## Figures and Tables

**Figure 1 entropy-21-00209-f001:**
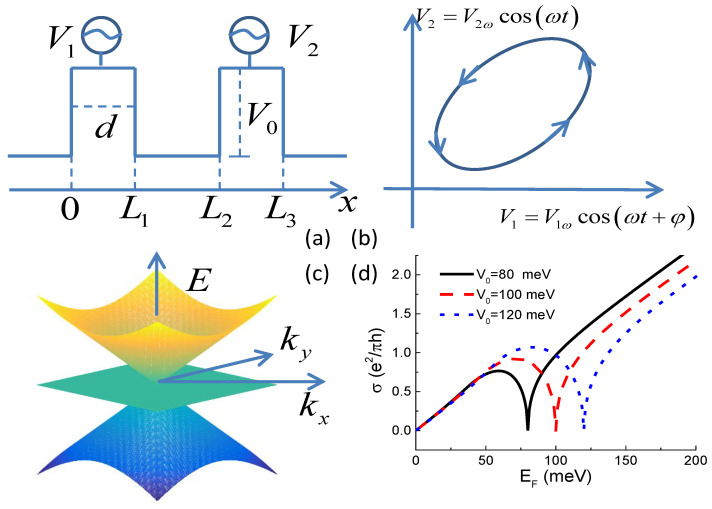
(**a**) schematics of the adiabatic quantum pump. Two time-dependent gate voltages with identical width *d* and equilibrium strength $V_{0}$ are applied to the conductor. Time variation of the two potentials $V_{1}$ and and $V_{2}$ is shown in panel (**b**). $V_{1}$ and $V_{2}$ have a phase difference giving rise to a looped trajectory after one driving period; (**c**) two-dimensional band structure of the pseudospin-1 Dirac–Weyl fermions with a flat band intersected two Dirac cones at the apexes; (**d**) conductivity of the pseudospin-1 Dirac–Weyl fermions measured by [54] $\sigma =\frac{e^{2}k_{F}d}{\pi h}\int_{-\pi /2}^{\pi /2}{\left|t(E_{F},\theta )\right|}^{2}\cos\theta d\theta$ in single-barrier tunneling junction as a function of the Fermi energy for three different values of barrier height $V_{0}$. $k_{F}=E_{F}/\hbar v_{g}$ is the Fermi wavevector and *t* is the transmission amplitude defined in Equation (Equation 5). It can be seen that higher barrier allowing larger conductivity occurs at the Dirac point $E_{F}=V_{0}$ and around $E_{F}=V_{0}/2$ (see the text).

**Figure 2 entropy-21-00209-f002:**
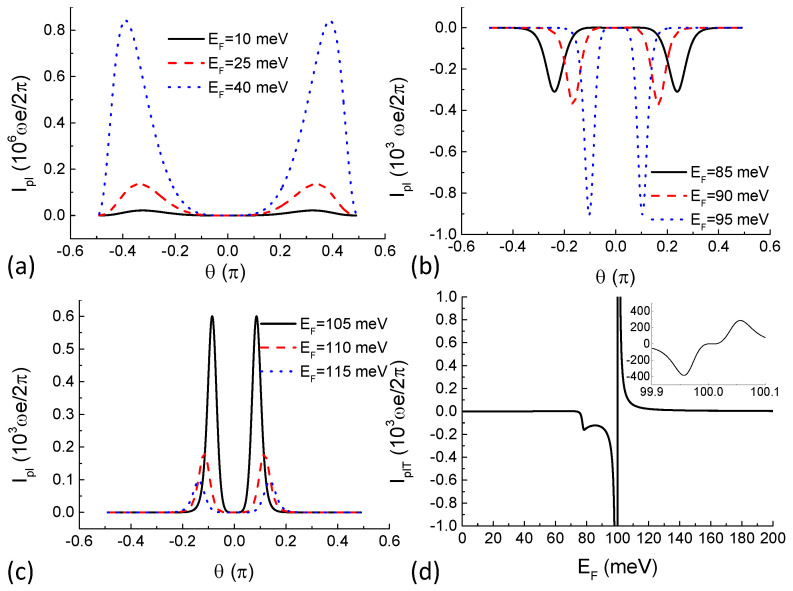
(**a**–**c**): angular dependence of the pumped for different Fermi energies with the driving phase difference φ fixed; (**d**) angle-averaged pumped current as a function of the Fermi energy. Its inset is the zoom-in close to the Dirac point to show that the large value of the pumped current does not diverge. Other parameters are $V_{0}=100$ meV, $V_{1\omega }=V_{2\omega }=0.1$ meV, d=5 nm, $L_{2}-L_{1}=10$ nm, and φ=π/2.

**Figure 3 entropy-21-00209-f003:**
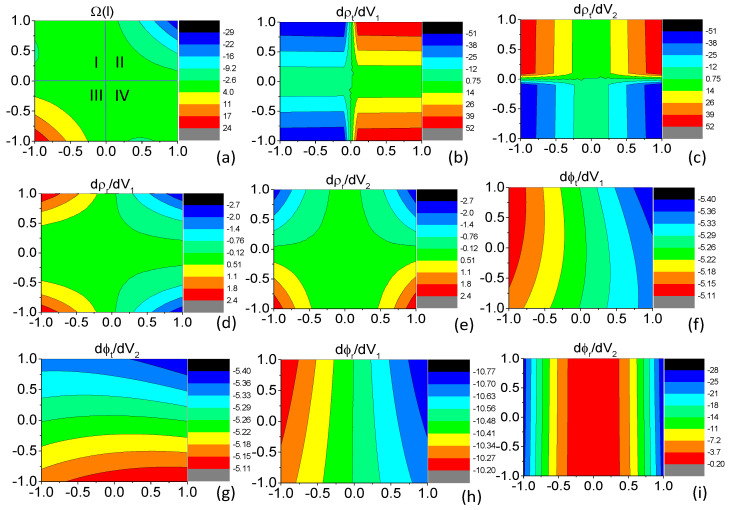
Contours of the Berry curvature $\Omega \left(l\right)$ and the eight derivatives on the right-hand side of Equation (Equation 10) in the $V_{1}$-$V_{2}$ parameter space. For all the subfigures, the horizonal and vertical axes are $V_{1}$ and $V_{2}$ in the unit of meV, respectively. The magnitudes of the contours are in the scale of (**a**) $10^{-7}$; (**b**) $10^{-4}$; (**c**) $10^{-4}$; (**d**) $10^{-5}$; (**e**) $10^{-5}$; (**f**) $10^{-2}$; (**g**) $10^{-2}$; (**h)**$10^{-2}$; (**i**) $10^{-2}$; and (**j**) $10^{-5}$, respectively. Other parameters are $V_{0}=100$ meV, d=5 nm, $L_{2}-L_{1}=10$ nm, $E_{F}=100$ meV, and θ=0.5 in radians. For convenience of discussion, the parameter space in the nine panels is divided into four blocks: I ($-1<V_{1}<0$ and $0<V_{2}<1$), II ($0<V_{1}<1$ and $0<V_{2}<1$), III ($-1<V_{1}<0$ and $-1<V_{2}<0$), and IV ($0<V_{1}<1$ and $-1<V_{2}<0$). The four blocks are illustrated in (**a**).

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
