# Peer review of "Quantum Pumping with Adiabatically Modulated Barriers in Three-Band Pseudospin-1 Dirac–Weyl Systems"

_entropy, 2019, doi:10.3390/e21020209_

Round 1

Reviewer 1 Report

In this paper the authors calculate the adiabatically pumped current in a system consisting of a two-dimensional non-interacting pseudospin-1 Dirac-Weyl slab in the presence of two time-dependent potential barriers. They use the Brouwer's scattering formalism. The main finding is that the sign of the current is opposite to the one expected from the time-dependent height of the barriers (if a turnstyle picture is considered). This is attributed to the super Klein tunnelling effect and to the presence of particle-hole symmetry.

The paper is clearly written and the results are original. The reference made to the concept of Berry phase, though, is misleading (see comments below). The paper deserves publication in Entropy once the remarks below are considered.

Remarks:

1) In section 3 the authors speak about to conductivity, while it should be conductance (as given by the Landauer-Buttiker formula through the transmission probability).

2) The local maximum of the conductance shown in Fig. 1(d) as a function of the Fermi energy looks like it is not close to half the barrier height (contrary to what it is stated in line 92), but much more to the right, as actually written below (line 94). This confusing statements should be avoided.

3) I do not understand the statement in line 96 "... the existence of the two maximum peaks ...". Fig. 1(d) shows only one maximum. This should be clarified.

4) Tick labels in the inter of Fig. 2(d) are not visible, too small.

5) Regarding section 4, the authors associate the positivity of the Berry phase with the height of the potential barrier. I do not see what is the significance of this, why it is interesting. The concept of Berry phase it is not needed in this discussion. This seems to me just a check that Eqs. (6) and (7) are correctly implemented. The definitions in Eqs. (10) and (11) are redundant (see Eq. (7) and discussion).

6) Labels and tick labels in Fig. 3 are too small.

Author Response

Dear Reviewer, 
Thank you for your comments on our manuscript (entropy-406166). Enclosed please find our revised manuscript for your second-round review. Below you find a summary of the changes made and a brief response to all recommendations and criticisms. Changes are boldfaced in the revised manuscript.

Thank you once again for your further consideration, and we are looking forward to hearing from you.

Sincerely yours,

Rui Zhu

[on behalf of all authors]

Reviewer 2 Report

The authors study the pumped current by adiabatically driving a double barrier structure in a pseudospin-1 Dirac Weyl fermion model.
They find that a sharp current direction reversal for special values of the parameters at the Dirac point of the band structure. The finding of such current reversal and the interpretation in terms of turnstile mechanism is very interesting. But before the manuscript is acceptable for publication a deep revision of the manuscript is required.
-When introducing the scattering matrix approach the authors do not explicitly write the boundary conditions of the wavefunctions and its derivative at the interface to derive the scattering coefficients. For clarity they should write them.
-The expression of the scattering coefficients are not explicitly given (nor in an appendix, nor in a supplementary material) so the results have only to be trusted. At least the main dependence on the angle and Fermi level should be clearly written in the main text.
-All the discussion on the consistency between the turnstile model and the Berry phase is very generic and is valid for any system with a Dirac cone, so what is characteristic of the Dirac Weyl fermion model and what is the difference from the graphene model (that one of the authors already treated)?
-Again after Eq (12) there is no comment on the dependence of \rho and \phi on the parameters V_1 and V_2 and the results have to be again only trusted.
-If the authors could related all the results of pag. 7 to the specificity of the band structure of the model would be very valuable and stress the difference from the monolayer graphene would be very valuable.

So the manuscript contains interesting results but an extensive revision is necessary to render the results transparent and to clarify the major differences of the case discussed by the graphene monolayer model.

Author Response

(The authors gave the same response as above.)

Reviewer 3 Report

the authors study the quantum pumping of current in pseudo-spin 1 fermion, the analysis interns of berry curvature and the turnstile model is useful and interesting to the readers. in general the presentation is reasonable and is acceptable after some minor language checking.the title of my copy seems to be wrong. i am not sure if it is the problem of the generation of the pdf or a problem of the orginal manuscript.

Author Response

Dear Reviewer, 
Thank you for your comments on our manuscript (entropy-406166). Enclosed please find our revised manuscript for your second-round review. Your comments about the manuscript are valuable for us to improve the work. The issues your raised are addressed as below. Thank you once again for your further consideration, and we are looking forward to hearing from you.   Sincerely yours, Rui Zhu [on behalf of all authors]

Round 2

Reviewer 2 Report

In the revised version the authors did took all the criticisms properly into account and tried to give a more convincing argument about the equivalence between the turnstile and the Berry phase model. The manuscript can now be accepted for publication.

Author Response

Thanks for your positive comments.